# Effectiveness of an *O*-Alkyl Hydroxamate in Dogs with Naturally Acquired Canine Leishmaniosis: An Exploratory Clinical Trial

**DOI:** 10.3390/ani12192700

**Published:** 2022-10-07

**Authors:** Victoriano Corpas-López, Victoriano Díaz-Sáez, Francisco Morillas-Márquez, Francisco Franco-Montalbán, Mónica Díaz-Gavilán, Julián López-Viota, Margarita López-Viota, José Antonio Gómez-Vidal, Joaquina Martín-Sánchez

**Affiliations:** 1Departamento de Parasitología, Facultad de Farmacia, Universidad de Granada, 18011 Granada, Spain; 2Departamento de Química Farmacéutica y Orgánica, Facultad de Farmacia, Universidad de Granada, 18011 Granada, Spain; 3Departamento de Farmacia y Tecnología Farmacéutica, Facultad de Farmacia, Universidad de Granada, 18011 Granada, Spain

**Keywords:** vorinostat derivatives, canine leishmaniosis, treatment, *O*-alkyl hydroxamates, histone deacetylase inhibitors, effectiveness, safety

## Abstract

**Simple Summary:**

Canine leishmaniosis is a challenge in veterinary medicine and no drug to date has achieved parasite clearance in dogs. We have recently found that Vorinostat derivatives (*O*-alkyl hydroxamates) are active against Leishmania infantum intracellular forms and effective in laboratory models of visceral leishmaniasis without producing toxicity. We designed a clinical trial with 18 dogs naturally infected with the parasite and demonstrated that our flagship compound, MTC-305, is superior to the current first-line treatment of canine leishmaniasis, meglumine antimoniate, at reducing the parasite numbers in target organs (bone marrow, lymph nodes and blood) and improving the dogs’ clinical state. MTC-305 was not toxic in dogs, unlike the standard treatment that causes gastrointestinal alterations. We believe that, despite the limitations of the present work, MTC-305 and other *O*-alkyl hydroxamates are promising drugs in the fight against this neglected disease.

**Abstract:**

Canine leishmaniosis is a challenge in veterinary medicine and no drug to date has achieved parasite clearance in dogs. Histone deacetylase inhibitors are a drug class widely used in cancer chemotherapy. We have successfully used *O*-alkyl hydroxamates (vorinostat derivatives) in the treatment of a laboratory model of visceral leishmaniasis without showing toxicity. In order to test the effectiveness of a particular compound, MTC-305, a parallel-group, randomized, single-centre, exploratory study was designed in naturally infected dogs. In this clinical trial, 18 dogs were allocated into 3 groups and were treated with either meglumine antimoniate (104 mg Sb^V^/kg), MTC-305 (3.75 mg/kg) or a combination of both using a lower MTC-305 dose (1.5 mg/kg) through a subcutaneous route for 2 treatment courses of 30 days, separated by a 30-day rest period. After treatment, a follow-up time of 4 months was established. Parasite burden in bone marrow, lymph node and peripheral blood were quantified through qPCR. Antibody titres were determined through an immunofluorescence antibody test, and cytokine expression values were calculated through RT-qPCR. Treatment safety was evaluated through the assessment of haematological and biochemical parameters in blood, weight, and gastrointestinal alterations. Assessment was carried out before, between and after treatment series. Treatment with MTC-305 was effective at reducing parasite burdens and improving the animals’ clinical picture. Dogs treated with this compound did not present significant toxicity signs. These results were superior to those obtained using the reference drug, meglumine antimoniate, in monotherapy. These results would support a broader clinical trial, optimised dosage, and an expanded follow-up stage to confirm the efficacy of this drug.

## 1. Introduction

Canine leishmaniosis (CanL) is a dog multisystemic disease caused by *Leishmania infantum* [1,2], a protozoan parasite that also causes visceral, mucosal and cutaneous leishmaniasis in humans across the world, including the Mediterranean basin, Middle East, China and Latin America, where it is endemic [3,4]. Dogs are the main reservoir of this widespread parasite and once infected they can be asymptomatic carriers or develop the disease, whereby, unlike in humans, cure is an uncommon event, since the dog’s immune system cannot produce an effective response to eliminate the parasite even under treatment. Since available drugs cannot eliminate the pathogen and *L. infantum* reaches deep organs such as the bone marrow, therapies focus on improving the animal’s state, strengthening the immune system, decreasing the ability to transmit the parasite to vectors and preventing clinical relapses, which are common once the treatment is over.

Even though pentavalent antimonials are still the first-line drugs for CanL [5], they exhibit high toxicity and the risk of resistance is high [6]. Allopurinol is usually administered in combination, alone or with antimonial therapy in order to prevent the occurrence of relapses [5]. Some drugs used in human visceral leishmaniasis, such as miltefosine and aminosidine, have been found efficacious against CanL in monotherapy or in combination with allopurinol; however, parasite elimination was not achieved and dogs suffered relapses at the end of the study [7].

Histone deacetylases (HDAC) inhibitors have proved effective in the treatment of cancer: several molecules such as vorinostat have been approved for the treatment of cutaneous T-cell lymphoma and they are showing promising activity in laboratory models of malaria [8]. Histone-modifying enzymes, such as HDAC, are essential for the modulation of chromatin structure, thus indirectly regulating gene expression in eukaryotic species. Their relevance seems even higher in trypanosomatid parasites, organisms that lack canonical transcription regulation.

We have recently demonstrated that *O*-alkyl hydroxamates (vorinostat derivatives) display potent and selective in vitro activity against *L. infantum* and are effective in a laboratory model of visceral leishmaniasis, both in monotherapy or in combination with meglumine antimoniate without displaying toxicity [9,10].

CanL is a veterinary practice challenge that is directly associated with public health, considering its high prevalence and closeness with humans [1,11]. The aim of this study was to evaluate the promising antileishmanial agent MTC-305 in dogs with naturally acquired CanL, in order to assess its effectiveness and safety profile in the dog and support the feasibility of a broader clinical trial. For that purpose, a randomized exploratory or pilot clinical trial was designed.

## 2. Material and Methods

### 2.1. Drugs

Meglumine antimoniate (Glucantime^®^) was purchased from Merial (Barcelona, Spain). MTC-305 was synthesised and nanoencapsulated as previously described [10].

### 2.2. Study Design

A randomized, parallel-group, single-centre, exploratory study with balanced randomisation (1:1:1 ratio) was designed and carried out in the facilities of an animal shelter (registry number ES189790000096) from September 2014 to February 2015. The animal shelter staff, the veterinarian and the researchers were kept blinded. Only the person administering the drug was not blinded.

### 2.3. Dog Enrolment

Eighteen privately owned dogs were enrolled in the clinical trial. All owners lived in Granada province, where leishmaniasis due to *L. infantum* is endemic and CanL prevalence is high [1]. Three inclusion criteria were established: two or more signs compatible with CanL, IFAT titre ≥ 40 and positive *L. infantum* qPCR. Exclusion criteria were as follows: pregnant and lactating bitches, dogs presenting with other infectious diseases, dogs that had recently received antileishmanial treatment (within 2 years) and dogs presenting with severe liver or kidney disease. Some measures were taken to protect the dogs from vectors: small-grid nets were placed in the facilities, dogs wore deltamethrin-impregnated collars Scalibor^®^ (MSD Animal Health, Kenilworth, NJ, USA) and a repellent-insecticide, Advantix^®^ (Bayer, Leverkusen, Germany), was applied in September 2014, July 2015 and October 2015. Dogs were checked for their vaccination status, and they were dewormed, treated for ectoparasites and analysed for other infections (i.e., ehrlichiosis) as per the animal shelter’s guidelines by the veterinary practitioner.

### 2.4. Treatment Allocation

Dogs were randomly allocated to the treatment groups: MTC-305 (*n* = 6), meglumine antimoniate (*n* = 6) or a combination of both (*n* = 6). This allocation was carried out using a concealed list of random numbers once a dog passed the baseline evaluation. The interventionist and the veterinary practitioner assessed treatment compliance. The shelter staff and the veterinarian monitored the dogs on a daily basis.

### 2.5. Drug Therapy Intervention

The intervention consisted of two 28-day treatment series (starting on day 0 and 60) with a 1-month rest between these treatments. MTC305 was administered at 3.75 mg/kg/day, subcutaneously; meglumine antimoniate (MA) was administered subcutaneously at 104 mg Sb^V^/kg/day; the combination group was administered MA (104 mg Sb^V^/kg/day) and MTC305 (1.5 mg/kg/day). A follow-up period of 4 months started at the end of the treatment. The dogs were assessed by the research team and the veterinarian 4 times: baseline evaluation (day 0), 30 days after the first treatment course (day 60), 30 days after the second treatment course (day 120) and at the end of the follow-up period (day 210). At every assessment, a complete clinical picture of the dogs was carried out including blood and parasitological tests.

### 2.6. Outcome Assessment: Drug Efficacy

A 50% reduction of the parasite load in bone marrow, lymph node and peripheral blood by day 120 was established as a marker of efficacy (primary outcome measure).

Other measures such as the clinical score and the immune status were evaluated (secondary outcome measures).

### 2.7. Outcome Assessment: Safety Evaluation

Weight, appearance and behaviour were recorded at every analysis point. Blood tests were performed in order to evaluate biochemical, enzymatic and haematological parameters. The following safety issues were categorised as toxicity events that would lead to the withdrawal of the dog from the study: kidney damage, liver damage, chronic vomiting, chronic diarrhoea.

### 2.8. Sample Collection and Processing

Cephalic venipuncture was used for peripheral blood collection. These samples were split into aliquots for qPCR (EDTA tubes), biochemical and haematological tests (heparin and EDTA tubes) and serum (clean Eppendorf tubes) for antibody determination. Bone marrow aspirates were taken via aspiration and stored in EDTA tubes for cell culture and qPCR. Lymph node samples were also obtained via popliteal ganglion aspiration and used for culture and qPCR. For parasite culture, samples were inoculated in culture tubes and incubated; the rest of the samples were stored at −20 °C.

### 2.9. Clinical State Evaluation

Clinical state was assessed at every time point through the scoring of clinical signs on a 0–3 scale (absence to severe), as previously reported [12]. Clinical score (CS) resulted from the addition of all sign scores.

### 2.10. Parasite Isolation

EMTM (agar blood solid phase) with RPMI-1640 medium as liquid phase and supplemented with 20% FBS was used to incubate bone marrow and lymph node aspirates. Culture tubes were incubated at 26 °C and sub-inoculated weekly until they were rejected after two months.

### 2.11. Parasite Load Quantitation

DNA was extracted from blood (200 µL), bone marrow samples (200 µL) and lymph node aliquots using a commercial kit (MasterPure^®^ Extraction Kit, Epicentre, Charlotte, NC, USA), as previously described [13]. Parasite burden in these tissues was quantified via qPCR, as previously described [13].

### 2.12. Cytokine Expression

RNA extraction, complementary DNA synthesis and cytokine expression analysis (interleukin 4 (IL4) and γ-interferon (IFNG)) was performed using a retrotranscriptase-quantitative PCR technique, as previously described [12].

### 2.13. Immunofluorescence Antibody Test

Immunofluorescence antibody test (IFAT) was carried out to measure *L. infantum*-specific antibodies in dog sera, as reported elsewhere [12].

### 2.14. Ethical Statement

Owners filled in an informed consent form and granted permission to house their pets in our facilities, to administer Glucantime^®^, MTC-305 or their combination to their dogs and to take samples as above. This clinical trial was approved by the Ethics Committee of the University of Granada and the Andalusian Ministry of Agriculture, in agreement with European law (Directive 2010/63/EU).

### 2.15. Statistical Analysis

Calculations on sample size estimated that three dogs were necessary per group in order to detect 80% parasitaemia reduction, 80% power and a 5% significance level. These estimations were based on our group’s interim data. Mann–Whitney and Kruskal–Wallis tests were used to find differences among antibody levels and cytokine expression values. Linear mixed models were used to evaluate parasite load and CS evolution using time, dog identity and treatment as independent variables. Dog identity was labeled as a random effect to account for dog variability.

## 3. Results

### 3.1. Participants and Participants Flow

Out of 118 dogs evaluated, 18 were recruited for the study and were randomly assigned to the study groups. All dogs received the treatment and completed the follow-up period. Dog age, sex, breed, IFAT titre and signs compatible with CanL during the selection stage are shown in Appendix A.

### 3.2. Efficacy of MTC305 and Its Combination

MTC305 treatment in monotherapy was effective in all six dogs, producing a marked parasite load decrease at day 120: 89.5% reduction in bone marrow (BM), 96% reduction in lymph node (LN) and 53% reduction in parasitaemia (PB). Consequently, their clinical score decreased in 5 out of 6 dogs at an average 60%. In the absence of drug pressure, parasite load continued to decline in some dogs, whereas it recidivated in others (dogs 7 and 10, day 210, Table 1). Antibody titres changed slightly during the analysis despite the overall parasite burden reduction, showing a mixed Th1/Th2 response, whereas IL4 expression decreased in five of the dogs. Although the immune response was mixed during the clinical trial, IgG2 increased in dogs that did not experience a relapse (Table 2 and Appendix A).

The combination was effective in 5/6 dogs: BM, LN and PB parasite loads decreased in more than 90% of these five dogs (day 120, Table 3); conversely, LN parasite load increased in dog 14. Clinical score decreased in all animals, including dog 14. The immune response progression was divergent in this group as well except for dog 13, which experienced a marked Th1 response increase through the increase of the IgG2/IgG1 and IFNG/IL4 ratios. At the end of the follow-up (day 210, Table 4), relapses occurred in two dogs for which treatment worked (dogs 16 and 18).

Conversely, meglumine antimoniate (MA) in monotherapy was considered effective in only one out of six dogs (Appendix A).

The regression analysis performed revealed that treatment with MTC-305 significantly reduced parasite load in BM (*p* < 0.005) and LN (*p* < 0.005). The analysis found association between treatment with MTC-305 and the clinical picture improvement (*p* < 0.05).

### 3.3. Treatment Safety

MTC-305 in monotherapy did not cause weight loss, diarrhoea or vomiting. The only haematological alteration found was a decrease in segmented neutrophils in the day 120 analysis that disappeared in the day 210 analysis, alteration shared with dogs treated with MA. Regarding biochemical alterations, lipase, amylase and urea increased in MTC-305-treated dogs without reaching the upper limit after treatment (day 120) but this alteration disappeared in the follow-up analysis.

The combination therapy seemed more aggressive despite the lower MTC-305 dose: there was a weight loss after the first treatment series in four dogs (day 60), but this alteration did not appear after the second treatment series. No vomiting or diarrhoea events were reported in this group and only some biochemical values increased in some of the dogs (amylase, urea and bilirubin) but these values were within the normal range.

MA treatment led to a weight loss >5% in four out of six dogs treated with this reference compound. Platelets and segmented neutrophils values decreased after the treatment with this drug. Other alterations included high amylase values in one dog and elevated bilirubin values in several animals in this group.

## 4. Discussion

The present study constitutes a pilot or exploratory clinical trial with the objectives of assessing the safety of an *O*-alkyl hydroxamate (MTC-305) in dogs with CanL and comparing its effectiveness with meglumine antimoniate (MA), the reference treatment in Europe [14], in order to find out if a larger clinical trial is feasible. This is usually the aim of the pilot, exploratory or proof of concept trials [15,16]. In addition to these objectives, we evaluated the effectiveness of a combination of MTC-305 and MA given its good results in mice. The number of dogs in this study was small but it was adequate in terms of statistical power based on previous data gathered by our group. Ideally, a study with a larger number of animals could be designed given the positive results of this exploratory study.

Inclusion criteria were not strict, but have been used in the past [17], and allowed a mixed dog cohort in the study that reflects the actual clinical picture of dogs with CanL, with a mixed population of oligosymptomatic and polisymptomatic animals (phase I to III according to [18]). Although the dog groups were mixed, the groups were homogenous regarding age, sex and clinical state.

Regarding intervention structure, dose and length, MTC-305 treatment was adjusted to the most common MA regime, and the dose was extrapolated from the one used in mice using body surface area approximation [19]. Antimonial therapy in dogs is usually administered in two treatment series in order to reduce the chance of relapse and treatment failure [20]. In the present study, we applied this structure allowing a 30-day interval period between series. The follow-up period was established at 4 months, under funding pressure. Some authors consider that a follow-up period should be at least 6 months long given that relapses can occur afterwards and antibody titres usually take longer than 3 months to change [21,22]. The results of this exploratory clinical trial could be used to design an optimized study with a larger number of dogs, an adjusted drug dose and a broader follow-up period.

Parasitaemia is usually evaluated in the monitoring of treatment effectiveness in dogs as it is less invasive than other tissues or organs [23]. Clinical score is usually employed too as it shows correlation with parasite load and antibody titres [24,25]. Bone marrow is a heavily parasitized organ in most dogs with CanL, which usually leads to changes in this tissue and subsequent alterations associated with splenomegaly [26]. Some studies have found high parasite load in this tissue regardless of their clinical state [27], and other authors have considered bone marrow the best biomarker to monitor the dog’s clinical picture [28] given its association with antibody titres and parasite loads in spleen. Therefore, the present study considered bone marrow parasite burden the primary outcome measure, whereas lymph node and peripheral blood were considered in dogs in which the parasite was absent in bone marrow. In our study, lymph node and particularly bone marrow parasite load, were associated with the clinical state.

Clinical score (CS) in our study was 6.9 on average (in the range 2–24), which is low compared to the maximum achievable score (67), but it is similar to other studies such as that of Proverbio et al., 2014 (mean CS 5.1 and the maximum CS found 30) [25]. We found the bone marrow to be the most parasitized tissue studied, followed by lymph node and peripheral blood, similar to other studies [29,30]. Regarding antibody titres, total IgG was very variable and IgG2 was larger than IgG1, similar to other studies [28]. IgG isotypes interpretation is not consistent across literature as some authors associate it with their clinical state [31], whereas other authors cannot find any correlation [32]. In the present work, IgG isotypes, particularly IgG1, showed a positive correlation with the CS. IFNG expression (associated with Th1 response) was higher than IL4 (linked to disease progression and Th2 response) in most dogs, confirming that Th1 response appeared higher in most individuals. However, it should be noticed that only two cytokines were evaluated in this study and other actors may have relevance in the immune response. The bone marrow parasite load found in this study showed good correlation with IgG1 and IL4.

MA was only effective in one treated dog in the present study; consequently, their immune response leaned towards Th2 in 4 dogs and towards Th1 in the remaining two. This therapeutic failure could be attached to the use of MA in monotherapy: 82% veterinary practitioners use antimonials in combination with allopurinol in Mediterranean Europe. However, our results with MA were poorer than those previously reported [33,34]. Drug resistance is one of the main issues in antileishmanial chemotherapy [35], and CanL is not exempt from this problem [6,36], which may have played a role in this treatment failure.

The effectiveness of MTC-305 has been demonstrated in the present study: treatment reduced BM, LN and PB parasite loads by more than 50% in all dogs. This effect did not seem associated with the animals’ previous state since the immune response pattern was similar among the groups (IgG2 titres were higher than IgG1 and IFNG expression was higher than IL4). Clinical score also decreased at the end of the treatment. Although the immune response only varied slightly, IL4 seemed to decrease during treatment and IgG2 increased, particularly in dogs that did not recidivate. After the follow-up period, a relapse was found in some of the dogs. An in vivo pharmacokinetic study of the drug would be very beneficial to improve the dosage regime of this drug in order to maintain drug pressure at an optimal level for longer periods.

A drug combination of MA and MTC-305 at a lower dose was effective in all but one dog. Similar to the MTC-305 treatment in monotherapy, the immune response did not change significantly, and its evolution was dissimilar. Drug combinations are widely use in antileishmanial chemotherapy in order to reduce toxicity, relapses and resistance generation. MA is usually employed in combination with allopurinol, a leishmaniostatic drug that is also administered in monotherapy in asymptomatic or stage 1 dogs. In this instance, MTC-305 dose was reduced in order to reduce toxicity and interaction risk with MA.

The different therapies were well tolerated overall. MA safety has been evaluated in the past [37,38], but in the present study, no hepatotoxicity, kidney or pancreatic toxicity were found, apart from increases in some parameters (creatitine, lipase and amylase) at different points. Diarrhoea and weight loss were found, a finding that has been reported previously [33]. Dogs treated with MTC-305 did not experience weight loss or gastrointestinal alterations. Some dogs showed slight biochemical alterations during the study that returned to normality in the following evaluation, but they were not indicative of kidney, hepatobiliary and pancreatic damage according to international guidelines [39,40,41,42].

Vorinostat, a MTC-305 parent compound, has been tested using preclinical models, showing little or no toxicity. It only showed gastrointestinal alterations in dogs and the NOAEL was established at 60 mg/kg. Cardiac alterations and mutagenesis have also been discarded in this compound [43]. MTC-305 has been previously assessed, showing no mutagenesis potential or activity over hERG channels [10].

No significant toxicity was recorded in dogs treated with the combination therapy. Four dogs suffered a 5% weight loss after the first treatment series, but they had recovered it by day 120. Although this combination was expected to be more aggressive, their safety profile was similar to dogs treated with MTC-305, recording some mild alterations that improved over time. The positive disease evolution in this group of dogs may have prevented other effects that were recorded in dogs treated with MA, which in turn suffered a disease deterioration.

Our results indicate that MTC-305, both in monotherapy and as a combination with MA, was more efficacious and safer than the reference antimonial treatment, in the context of a limited pilot exploratory clinical trial. A broader study that includes an optimised dosing through pharmacokinetic profiling, a large number of animals and a broader follow-up would be ideal to reinforce these findings.

## 5. Conclusions

The effectiveness of the vorinostat derivative MTC-305 has been assessed in a small cohort of dogs involving a deep analysis of disease biomarkers. The effectiveness of this drug in this incurable animal model, superior to the reference treatment, makes a larger clinical trial feasible, including an optimised dosage and a broader follow-up taking advantage of the results of this pilot or exploratory clinical trial. In addition, safety and efficacy in canine leishmaniasis puts forward the idea of its use in human leishmaniases.

## Figures and Tables

**Table 1 animals-12-02700-t001:** Parasite load and the clinical state during the experiment in MTC-305-treated dogs. BM, bone marrow (parasites/µL); LN, lymph node (parasites/µg DNA); PB, parasitaemia (parasites/100 µL); CS, clinical score.

		Parasite Burden or CS			
Dog	Sample	Day 0	60	120	210
7	BM	0	0	0	0
	LN	1.04	0	0	1.51
	PB	0.01	0	0	0
	CS	3	2	2	2
8	BM	82.4	3.16	19.2	4.61
	LN	270.2	33.2	2.61	43.7
	PB	0.004	0.004	0.007	0.005
	CS	10	10	13	20
9	BM	0.09	0	0	0
	LN	0	0	0	0
	PB	0.007	0.001	0	0
	CS	7	5	1	2
10	BM	0.013	0.004	0.003	0.011
	LN	0.83	0	0	0.26
	PB	0.01	0	0.003	0.01
	CS	2	1	1	1
11	BM	17.2	1.07	0.91	0.11
	LN	302.3	4.91	0.03	0.1
	PB	0.001	0.010	0.031	0.0005
	CS	14	12	4	4
12	BM	0.050	0	0	0.008
	LN	2.78	1.12	0.48	5.34
	PB	0.0025	0.0002	0.0029	0.0158
	CS	14	14	6	8

**Table 2 animals-12-02700-t002:** Antibody titres and cytokine expression during the experiment in MTC-305-treated dogs. tIgG, total IgG; IgG1, IgG1 subclass; IgG2, IgG2 subclass; IFNG, gamma interferon; IL4, interleukin 4.

		Titre or Expression Level			
Dog	Analysis	Day 0	60	120	210
7	tIgG	40	20	40	40
	IgG1	0	0	0	0
	IgG2	0	20	80	40
	IFNG	27.9	0.31	1.20	0.13
	IL4	0.40	0.07	0.03	1.80
8	tIgG	640	1280	1280	1280
	IgG1	0	0	0	40
	IgG2	1280	1280	1280	1280
	IFNG	7.78	19.7	21.6	17.8
	IL4	0.03	0.25	0.03	0.33
9	tIgG	1280	1280	1280	1280
	IgG1	80	80	80	40
	IgG2	1280	1280	1280	1280
	IFNG	0.64	0.65	13.5	3.89
	IL4	0.03	0.32	2.20	1.51
10	tIgG	40	0	40	40
	IgG1	0	80	20	0
	IgG2	40	80	80	40
	IFNG	47.8	44.0	0.8	78.2
	IL4	0.16	0.55	0.03	0.25
11	tIgG	80	40	1280	1280
	IgG1	0	0	80	80
	IgG2	80	80	320	1280
	IFNG	18.9	1.56	9.06	9.92
	IL4	0.03	0.03	0.03	0.07
12	tIgG	160	160	80	160
	IgG1	40	80	40	0
	IgG2	320	320	80	320
	IFNG	30.06	1.41	1.95	1.83
	IL4	1.55	0.34	0.49	0.03

**Table 3 animals-12-02700-t003:** Parasite load and the clinical state during the experiment in combination-treated dogs. BM, bone marrow (parasites/µL); LN, lymph node (parasites/µg DNA); PB, parasitaemia (parasites/100 µL); CS, clinical score.

		Parasite Burden or CS			
Dog	Sample	Day 0	60	120	210
13	BM	172.1	0	0	0.1
	LN	409.06	0.03	0.003	16.93
	PB	0.004	0.005	0	0
	CS	23	8	4	5
14	BM	0	0	0	0
	LN	0	1.69	0.18	0.60
	PB	0.001	0	0.001	0
	CS	4	2	2	2
15	BM	0	0	0	0
	LN	100.3	3.57	1.52	0.21
	PB	0.003	0.003	0.002	0
	CS	4	5	3	3
16	BM	0.001	0.0004	0	0.0069
	LN	1.98	0.00	0.02	5.11
	PB	0.09	0	0.01	0.10
	CS	2	1	1	1
17	BM	0.015	0.009	0.003	0.002
	LN	4.19	0	0	0
	PB	0.002	0	0.04	0.01
	CS	4	5	0	1
18	BM	0.0191	0	0	0.055523
	LN	0	0	0	0
	PB	0.07	0	0.024	0.013
	CS	2	0	0	0

**Table 4 animals-12-02700-t004:** Antibody titres and cytokine expression during the experiment in combination-treated dogs. tIgG, total IgG; IgG1, IgG1 subclass; IgG2, IgG2 subclass; IFNG, gamma interferon; IL4, interleukin 4.

		Titre or Expression Level			
Dog	Analysis	Day 0	60	120	210
13	tIgG	1280	1280	1280	1280
	IgG1	640	320	80	320
	IgG2	1280	1280	1280	1280
	IFNG	0.68	5.28	7.62	4.69
	IL4	0.30	1.42	0.50	0.68
14	tIgG	160	80	80	80
	IgG1	20	40	40	40
	IgG2	320	80	80	320
	IFNG	0.90	4.76	1.35	5.31
	IL4	1.34	0.03	0.03	0.03
15	tIgG	80	0	0	0
	IgG1	0	0	0	0
	IgG2	20	20	0	0
	IFNG	0.57	0.13	1.25	0.09
	IL4	0.03	0.69	0.03	0.03
16	tIgG	160	40	40	80
	IgG1	0	20	0	0
	IgG2	80	320	40	80
	IFNG	59.71	6.36	1.69	19.70
	IL4	0.37	0.49	0.38	1.75
17	tIgG	160	80	80	40
	IgG1	20	80	40	0
	IgG2	320	320	40	80
	IFNG	0.13	0.27	1.74	0.13
	IL4	1.08	0.03	0.15	0.07
18	tIgG	160	20	20	20
	IgG1	20	0	0	0
	IgG2	40	40	0	0
	IFNG	2.03	1.56	1.75	0.27
	IL4	1.37	0.32	0.03	0.03

## Data Availability

The data presented in this study are available on request from the corresponding author.

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
