# Peer review of "Effectiveness of an O-Alkyl Hydroxamate in Dogs with Naturally Acquired Canine Leishmaniosis: An Exploratory Clinical Trial"

_animals, 2022, doi:10.3390/ani12192700_

Round 1
Reviewer 1 Report
The manuscript reports results about a clinical study to evaluate the efficacy of the drug O-Alkyl hydroxamate in the treatment of canine visceral leishmaniasis. The study reports new results for the treatment of canine visceral leishmaniasis, but has some limitations in the methodology section and data analysis. In this format, the manuscript is not of interest to readers. Follows comments:
1) The number of animals in each experimental group is small;
2) The phenotypic characteristics of the animals are not reported (age, sex, breed, etc..), so they group aren´t homogeneous;
3) The health status of the animals is not reported (vaccination, deworming, ectoparasites and co-infections with other hemoparasites), the authors only report, in the line 88, “ that dogs with concomitant infections were excluded”, very vague;
4) The study design isn´t a case control;
5) The analyzes are not performed by means or median of each variable by group;
6) Tables must be replaced by graphs with the longitudinal monitoring of variables by treatment group;
Author Response
Dear reviewer 1,
Thanks for your comments. I believe we have improved the manuscript according to your useful suggestions, now clarified in the text. Changes are stated below:
- It is true that the number of animals is not ideal, that’s why this study was envisioned as a exploratory pilot clinical trial, also known as Proof of Concept. The idea behind this type of studies is to provide sufficient evidence to justify a bigger clinical trial. We did not have the funding for such study when this trial was planned and we may not have it in the future, but publishing these results could be of use for other groups or companies willing to investigate this drug. As stated in the Statistical analysis section, and based on previous interim data, to achieve a statistical power of 80% at a 95% confidence, a sample of just 3 dogs would have been necessary. Therefore, our experiment based on 6 dogs would be enough to find statistical difference between treated groups, and we actually found them using a general linear model that takes into account the variability of dogs (dog’s identity is included as a random effect). This is important for the reader to understand and it is explained throughout the paper (particularly in the introduction, the first paragraph of the discussion and the abstract and conclusion), but we have expanded these sentences now.
- We have included a new table (supplementary table 3) that shows the different characteristics of each dog. Groups were assigned randomly but they are homogenous regarding the sex, age and clinical state of the animals (Chi square test, p >0.1). The statistical analysis includes the dogs’ variability as state above and we believe that this variability reflects the clinical picture in the field.
- We have expanded the health status of the animals in the methodology section. Dogs were routinely checked by a veterinary practitioner that tests the dogs for other diseases (ehrlichiosis, filariasis, other bacterial or viral infections) and routinely dewormed and treated against ectoparasites liker other dogs in the animal shelter. Their vaccination status and clinical state was checked upon arrival to the facilities by the vet.
- I understand that you are highlighting the absence of a control (placebo or untreated) group. The study was initially designed to have a control group, but we decided that keeping a group of dogs untreated was not ethically acceptable and decided to only have treated groups. However, as seen in the results, treatment with the reference control (meglumine antimoniate) was not very effective, excluding placebo effect from our results.
- The general linear model analysis uses linear regression which is based on ANOVA test, therefore it uses parametric statistics measures such as the mean.
- We have now included graphs (supplementary figures S1 S2, S3 and S4) depicting the parasite loads and clinical score in the three target tissues. Graphs were not very informative for antibody titres and cytokine expression and therefore we have not included them. We believe that tables are very informative of the individual state and we have not removed them.
Reviewer 2 Report
the tested subjects are small in number. the follow up is too short to draw conclusions
Author Response
Dear reviewer 2,
Thanks for your comment. It is true that the number of animals is not ideal, that’s why this study was envisioned as a pilot clinical trial, also known as Proof of Concept. The idea behind this type of studies is to provide sufficient evidence to justify a bigger clinical trial. We did not have the funding for such study when this trial was planned and we may not have it in the future, but publishing these results could be of use for other groups or companies willing to investigate this drug. As stated in the Statistical analysis section, and based on previous interim data, to achieve a statistical power of 80% at a 95% confidence, a sample of just 3 dogs would have been necessary. Therefore, our experiment based on 6 dogs would be enough to find statistical difference between treated groups, and we actually found them using a general linear model that takes into account the variability of dogs (dog’s identity is included as a random effect). The follow-up period is a funding limitation of this study and this is mentioned in the paper from line 255. Ideally, a CanL study should have 1 year follow-up or at least 6 months. Given the nature of this study, a pilot or Proof of Concept clinical trial, the objective of this work was to assess the feasibility of a bigger clinical trial so that our group or other researchers can pursue a larger number of dogs, an optimised drug treatment schedule/dose and a longer follow-up period. This is important for the reader to understand and it is explained throughout the paper (particularly in the introduction and the first paragraph of the discussion, abstract and conclusion), but we have expanded these sentences now. I hope you can understand the nature of this paper and the importance of publishing these results.
Reviewer 3 Report
I would like to thank you for the opportunity to review this work. The identification of new lines of treatment for canine leishmaniasis is certainly of great interest and deserves to be considered for publication. There are some aspects that deserve clarification before considering the work for final approval. In particular, although the limitation imposed by financial availability is understandable, I am convinced that this is a fundamental limitation of the study. As reported by the Authors, the period usually suggested for follow-up after leishmanicidal treatment is at least 6 months, and this timing is of paramount importance to check for the possibility of recurrence (critical in case of parasitological failure to heal, as is the case in canine lesihmaniasis).
Throughout the article, when referring to the dog, I suggest replacing the term leishmaniasis with leishmaniosis because it is preferred in the latter period.
Line 86 "IFAT>40" replace with greater than or equal to 1:40. Granada being an endemic area, the Authors should explain why they used a cut-off of less than 1:80, which is the one classically used in all clinical trials conducted in endemic areas. The risk of enrolling patients who do not have clinical forms referable to leishmaniasis results in high and the results may have been influenced by this enrollment criterion.
Line 131 "evening" The authors mention serum, but tubes with which serum can be obtained are not mentioned above.
Line 134-135 Were bone marrow and lymph node samples dedicated to cultures also stored at -20°C?
Table 3 Dog 14 PB and CS are reversed.
Line 313-314 "Weight loss and diarrhea were noticed, in agreement with the fact that antimonial treatment usually leads to gastrointestinal alterations as well" Insert bibliographic reference.
Author Response
Dear reviewer 3,
Thanks for your insightful comments. I believe we have enhanced the text according to your useful suggestions, now clarified in the text. Changes are stated below:
- The follow-up period is a limitation of this study (due to funding) and this is mentioned in the paper from line 255. Ideally, a CanL study should have 1 year follow-up or at least 6 months. Given the nature of this study, a pilot or Proof of Concept clinical trial, the objective of this work was to assess the feasibility of a bigger clinical trial so that our group or other researchers can pursue a larger number of dogs, an optimised drug treatment schedule/dose and a longer follow-up period. This is shown in the first paragraph of the discussion and it’s very important for the reader to understand, therefore we have expanded this paragraph and mentioned this limitation in abstract and conclusion.
- The term “canine leishmaniasis” has been changed to “canine leishmaniosis”.
- Although the study area is endemic and an IFAT value of 80 is recommended, we established an IFAT=40 because we also screened the dogs using qPCR in blood. Therefore, the criteria for selection was IFAT=40, positive qPCR (blood) and two signs compatible with CanL. Even when the criteria was established in 40, all dogs selected had an IFAT titre >=80 (supplementary table 3).
- In line 131, tubes for serum are now specified.
- In lines 134-135, culture samples are now detailed. Bone marrow and lymph node samples were not frozen, but incubated in culture tubes.
- Table 3 has been amended, thanks for noticing.
- A reference has been added. Noli and Auxilia (2005) state that In dogs, the most commonly observed adverse effects associated with meglumine antimoniate are apathy, anorexia, vomiting, diarrhoea, and pain at the site of injection.